# On How Chronic Conditions Affect the Patient-AI Interaction: A Literature Review

**DOI:** 10.3390/healthcare8030313

**Published:** 2020-09-01

**Authors:** Mohammed Tahri Sqalli, Dena Al-Thani

**Affiliations:** Information and Computing Technology Division, College of Science and Engineering, Hamad Bin Khalifa University, Qatar Foundation, Doha P.O. Box 34110, Qatar; mtahrisqalli@hbku.edu.qa

**Keywords:** persuasive technology, health coaching, Human-AI Interaction, artificial intelligence, wellbeing

## Abstract

*Background*: Across the globe, managing chronic diseases has been recognized as a challenge for patients and healthcare providers. The state of the art in managing chronic conditions requires not only responding to the clinical needs of the patient, but also guaranteeing a comfortable state of wellbeing for them, despite living with the disease. This demands mutual effort between the patient and the physician in constantly collecting data, monitoring, and understanding the disease. The advent of artificial intelligence has made this process easier. However, studies have rarely attempted to analyze how the different artificial intelligence based health coaching systems are used to manage different types of chronic conditions. *Objective*: Throughout this grounded theory literature review, we aim to provide an overview for the features that characterize artificial intelligence based health coaching systems used by patients with chronic diseases. *Methods*: During our search and paper selection process process, we use three bibliographic libraries (PubMed, IEEE Xplore, and ACM Digital Library). Using the grounded theory, we extract overarching themes for the artificial intelligence based health coaching systems. These systems are then classified according to their role, platform, type of interaction with the patient, as well as targeted chronic conditions. Of 869 citations retrieved, 31 unique studies are included in this review. *Results*: The included studies assess 14 different chronic conditions. Common roles for AI-based health coaching systems are: developing adherence, informing, motivating, reminding, preventing, building a care network, and entertaining. Health coaching systems combine the aforementioned roles to cater to the needs of the patients. The combinations of these roles differ between multilateral, unilateral, opposing bilateral, complementing bilateral, one-role-missing, and the blurred role combinations. *Conclusion:* Clinical solutions and research related to artificial intelligence based health coaching systems are very limited. Clear guidelines to help develop artificial intelligence-based health coaching systems are still blurred. This grounded theory literature review attempted to shed the light on the research and development requirements for an effective health coaching system intended for patients with chronic conditions. Researchers are recommended to use this review to identify the most suitable role combination for an effective health coaching system development.

## 1. Introduction

Chronic diseases are considered among the most widespread and cost exhaustive health conditions around the world [1]. The World Health Organization (WHO) estimates that chronic diseases are the direct cause of more than 90% of morbidity and mortality rates in the upper and middle income countries [2]. As opposed to acute diseases, which have very distinct symptoms and a very short time-span, chronic diseases are permanent. The permanent nature of those conditions makes them strain the costs of treatment. The United States claims that it spends around $1.65 trillion on these conditions alone, which is around 75% of the annual yearly spending on healthcare for the nation ([3], p. 3). For that matter, technology is seeking innovative ways to find a mitigating solution to this urgent global threat [2]. Health and wellness is being integrated at the center of IoT and artificial intelligence (AI) technology [4]. The latest advancements in data analytics, cloud computing and big data are at the backbone of modern and digitalized healthcare systems ([5], p. 1) ([6], p. 3) [7]. The state of the art in managing chronic conditions focuses on providing patients with the necessary tools and trainings to promote their wellbeing. These tools and trainings are known as health coaching [8]. With the advent and affordability of intelligent technologies, enabling the health coaching process leads the patients to focus on their wellbeing despite the obstacle of living with a chronic condition ([4], p. 2). In recent years, several design transitions in the healthcare system’ s modes of operation to promote patients’ wellbeing emerged.

Healthcare systems around the world have adapted their modes of operation to suit the needs of patients coping with those diseases in order to effectively manage these growing conditions [8,9]. Therefore, these adaptations are reflected on the final design of health coaching systems (HCSs). Three main healthcare operational shifts are identified ([10], p. 22):The transition from a patriarchic to a collaborative way of managing the disease: The classic standard of care relies on a model where the physician has full authority and power over the treatment of the patient. Hence, the patients do not participate in curating the parameters of their treatment at all. The new operational shift provides more control and personalization for patients in terms of the way they manage their condition.The transition from an episodic to a continuous monitoring cycle for the patients’ health indicators. This aspect refers to the number of times a follow-up with the physician is held. Unlike for acute diseases, where follow-ups are rarely held, chronic conditions require a continuous monitoring process. Thanks to the advent of artificial intelligence and thanks to the affordability of sensors, patients’ indicators are measured around the clock for an efficient health and wellbeing management.The transition from prioritizing volume to prioritizing value for the healthcare provided: The focus for healthcare management has shifted from treating acute illnesses in a massive way, to more time allocated to understanding the patient. Value is added to the patients’ experience of managing their chronic disease through the use of connected health, tele-medicine, and tele-presence robotics [11,12,13,14]. This is achieved thanks to the availability and use of artificial intelligent systems that analyze patient’s data being collected through medical sensors. The end goal of using these technologies is to increase in the wellbeing of patients, and make the management process effective and seamless.

These three shifts in the healthcare system’s mode of operation have profound implications on how artificial intelligence-based health coaching systems are designed ([10], p. 23). The design of these systems is based on the founding idea that it needs to provide both coaching to manage the chronic condition of patients and, at the same time, promote the general wellbeing of the user. Moore sees these operational shifts as accomplishing cohesively dual objectives, mainly sustaining the life of the patient and promoting a good quality of this life. An example of successful adoption of these operational shifts is in the domain of caring for the wellbeing of patients with demantia and Alzhimer’s diseases [14]. Koutentakis et al. designed socially assistive robots for alzheimer’s disease patients and for their caregivers. The primary goal for these health coaching robots is the improvement of the wellbeing of the patients and their caregivers. Other examples are forthcoming in the analysis section (Section 5).

In this work, we perform a literature review using the Grounded Theory in Literature Review (GTLR) methodology [15]. We aim to investigate, from the lens of a designer, the patient-AI interaction paradigms between patients coping with a specific chronic condition and artificial intelligence systems that help patients manage it. Therefore, we focus on how the nature of the chronic condition impacts the software architecture of those systems. This is to explore and understand the nuances in the design of different artificial intelligence based health coaching systems. We attempt to correlate these nuances to the nature of the chronic disease the health coaching system addresses. We start by specifying the aim of this review. We also specify the motivation behind performing it. We then identify the materials and methods. Subsequently, we discuss the results of the search, the selection criteria, and finally the studies included. We then perform the analysis after selecting the studies, and finally discuss the results of the analysis. We conclude by outlining the limitations and future works.

## 2. Aim and Motivation

The use of e-health technologies in managing chronic diseases has increased. Moreover, healthcare providers understand the essential role of the patient’s feedback ([9,10], p. 19). The feedback that is generated from patients is essential in understanding their needs to eventually guarantee their wellbeing while they manage their disease. Khan et al. [16] attribute the increase of e-health solutions to the availability and abundance of sensors. These sensors enable the measurement of different aspects of the human health. The majority of the current systems harness the data collected by those health sensors using AI as a tool to better process patients information. AI is also incorporated in those health systems in order to provide a better quality of care. Therefore, it is essential to understand the applications, and the resulting consequences of incorporating AI in the medical sector. Additionally, understanding the risks of this digital strategy allows for the proper deployment of AI systems in health institutions. As a foundation for our analysis, we build on Amershi et al.’s analysis. We also take Nunes et al.’s work as a starting point. In both works, the authors suggest a way to build on their systematic and grounded theory literature reviews. Our motivation is to build on the authors’ work by contributing with a review on the use of AI system to provide health coaching for patients with chronic diseases. This work constitutes an extension to what Nunes et al. and Amershi et al. introduced, in the sense that it covers an update to the time-frame in which the two previous reviews were done. This work maintains the same analysis question. However, it directs the analysis process towards artificial intelligence based health coaching systems instead of rule-based systems. The question this work aims to answer is to investigate the criteria taken into consideration by the designers of artificial intelligence based health coaching systems during the design process. The fundamental research question that we aim to explore is:What is the influence of different types of chronic diseases on the design choices for AI-based health coaching systems for patients?

## 3. Materials and Methods

We perform a literature review using the Grouded Theory in Literature Review (GTLR) methodology in order to reach the aforementioned aim [15]. A GTLR is defined as a preliminary exploration of the available research body in order to extract the full theoretical value out of a well-chosen set of published studies. The review is guided by a systematic method. This method applies grounded theory to enhance the process of performing the search, selecting the titles and ultimately analyzing the studies to be included in the review. GTLR takes the data and conclusions from the research papers as empirical input that needs to be coded and contrasted.

### 3.1. Search Strategy

#### 3.1.1. Search Sources

The studies in the literature review are collected over the first half of the month of June 2020 from the following three electronic libraries: Pubmed, IEEE Xplore, and ACM Digital Library. Numerous papers from key academic venues according to the Scopus index for academic journals and conferences are included. Reference lists of the selected publications are checked for additional studies that are relevant to the topic (Backward reference list checking). Moreover, additional relevant studies are checked using the “Cited by” functionality available in the above-mentioned databases (Forward reference list checking).

#### 3.1.2. Search Terms

Search terms used in the current review are curated by taking into consideration three essential elements: target population (e.g., chronic condition, chronic diseases), healthcare intervention (e.g., health coaching, health care, health management), and technological solution (e.g., machine learning, artificial intelligence). The search terms are derived from the few studies that are available in the literature. These studies address e-health systems as a solution for self managing chronic conditions [17,18]. The search terms used are listed in the Appendix A.

### 3.2. Study Eligibility Criteria

In order for studies to satisfy the eligibility criteria, they need to convey primary research findings regarding artificial intelligence-based health coaching systems for a specific chronic condition. The review also includes articles that describe rule-based intelligent systems under one condition due to the novelty of the field, and due to the limited number of research articles pertaining to this area of research. The condition is that the system’s main goal it to serve for health coaching patients manage their chronic condition. Therefore, this review focuses on AI-based systems that coach patients to be more independent in managing their chronic condition. These systems may work on the following platforms: stand-alone software, web browser, specialized hardware dedicated for a specific chronic condition (like glucometers, heart monitors etc.), serious games, SMS based systems, and mobile applications. The review also includes studies containing health coaching systems that are designed to be used by both the patients and the coaches (like physicians, nurses, health coaches, and caregivers). Studies about AI-based health coaching systems whose main target audience is not coping with a chronic condition or chronic disease are excluded. The review includes peer-reviewed articles, dissertations, conference proceedings, and reports. However, it excludes reviews, conference abstracts, proposals, and editorials. Included articles need to be published starting from the year 2012. They also need to be written in the English language to be considered for selection in the literature review. No restrictions are imposed with regards to the type of the chronic condition, input and output modality, study design, study setting, measured outcome, or country of publication.

### 3.3. Study Selection

This review follows two major steps in selecting the studies. During the first step, two reviewers independently screen the titles as well as the abstracts of all initially retrieved studies. In the second step, the same reviewers study individually the full texts of the articles included from the first step. Any dis-accord between both reviewers is resolved through a discussion involving both reviewers about the concerned research paper. Inter-coder agreement between both reviewers is assessed while using Cohen’s kappa [19]. The ratios were 0.86 and 0.89 in the first and second step respectively across the selection process. This indicates very good agreement [19].

### 3.4. Data Extraction

In order to perform a systematic and precise data extraction, the reviewers developed a data extraction spreadsheet and tested it with few studies before using it. The data extraction step follows a similar process to the study selection one. In this step, the two reviewers independently perform the data extraction. Once disagreement occurs, it is collaboratively resolved and ultimately agreed upon. Inter-coder agreement between the reviewers was good (Cohen’s kappa = 0.85).

### 3.5. Study Quality Assessment

Grounded theory based literature reviews differ from systematic reviews. The difference lies in reviewing wider topics and including studies with varying study designs [15]. Thus, grounded theory-based reviews do not emphasize on the quality assessment of the included studies [15]. Consequently, the authors do not perform a formal assessment to the quality of the included studies.

### 3.6. Data Synthesis

We refer to a narrative approach in the data synthesis stage. We classify the health coaching systems according to the chronic condition addressed, the patient profiles, and the technical solutions. We also identify the duration of the study and the number of participants involved. Accordingly, we adopt the appropriate taxonomy published in the literature by Nunes et al. Characteristics of the studies and the population are summarised in Table 1. These characteristics are also described narratively. Subsequently, a description of the general characteristics of the health coaching systems is also presented.

## 4. Results

### 4.1. Search Results

Three bibliographic databases are searched to result in a total of 869 citations (PubMed *n* = 311, IEEE Xplore *n* = 185, ACM Digital Library *n* = 373). After removing 376 duplicates, 493 unique citations remain. After reviewing their titles and abstracts, 419 citations are excluded (whereby *n* = 277 are irrelevant, *n* = 18 have inadequate population, *n* = 43 have irrelevant intervention, *n* = 27 use irrelevant technology, and *n* = 34 have inadequate type of publication). By entirely reading the full text of the 74 remaining publications, 46 publications are excluded (whereby *n* = 4 discuss sports coaching, *n* = 24 solely discuss machine learning models for prediction or classification of diseases (without the use of a health coaching system for the patient), *n* = 13 are purely medical articles, and *n* = 5 articles compare public health systems between countries). 28 articles are therefore included. The included articles satisfy the study eligibility criteria detailed in Section 3.2 (i.e., studies that involve human subjects to test the effectiveness of a health coaching system for patients with a specific chronic disease). Two additional titles are identified from backward reference list checking. In sum, 31 publications are included in the synthesis. Table 1 summarizes the full list of included studies along with a summary of each study.

### 4.2. Description of the Included Studies

Table 2 provides an overall description of the included studies. Both quasi-experimental and survey study designs are used with a percentage of 45% and 41.9% respectively. Approximately two thirds of the studies are published in conference venues (*n* = 21, 67%). Although studies are authored in 11 different countries, more than one-third of the studies, about 38%, are authored in the USA (*n* = 12). About half of the studies are published between 2019 and 2018 (*n* = 15, 48%). The sample size is less than 50 participants in 19 studies and between 50 and 375 participants in 12 studies. The mean sample size is 54.17. The sample size ranges between 5 and 375 participants per study. The age of participants is only reported in 12 studies, whereby the mean age is 34.6 years. In the included studies, the recruited participants are either from a clinical setting (*n* = 9), a community setting (*n* = 17), or an academic setting (*n* = 5). Throughout the experiments conducted, the studies attempt to measure or quantify four types of outcomes. The effectiveness (*n* = 17), the usability (*n* = 5), the acceptability (*n* = 5) and the adoption (*n* = 4) of the artificial intelligence based health coaching system.

### 4.3. Axial Coding of the Elements Leading Towards an Innovative HCS

The process of performing axial coding on the included studies enables a detailed analysis of the different AI-based HCS types along with their roles. Figure 1 summarizes the axial coding performed in the context of artificial intelligence-based health coaching systems. The axial coding process starts with the causal conditions that either encourage patients with chronic conditions to either seek or refrain from referring to health coaching in general, and to AI-based HCSs in specific. Three major elements are key contributors to the causal conditions [47,48]: (1) The type of the chronic condition, (2) the cultural context of the patient coping with the chronic condition, and most importantly (3) the socio-economic situation of the patient. Patients at a low socio-economic context usually do not own the luxury to adopt an AI-based health coaching system to manage their condition [47,48]. Once causal conditions are defined, it is important to understand what are the repercussions of these conditions on the state of the patient dealing with them. Feelings of helplessness, powerlessness, and lack of control have been identified by several included studies among the patients [22,29,34,51]. Patient profiles are detailed more in-depth in Table 1. Moreover, the context in which the chronic condition is framed plays a key role in searching for strategies that are needed to cope with it. Under this context, four core frames are noteworthy: the sensation of pain that the condition brings to the patient, its frequency, its intensity, along with its duration. The strategies to find solutions to the health coaching problem are summarized in seven roles that the system plays to support the patients manage their chronic conditions. These roles come as a direct response to the problems identified. The roles are: developing adherence, informing, motivating, reminding, preventing, building a care network, and finally entertaining. Each role is discussed in details in Section 5.1. The final elements in the axial coding analysis are the consequences after the implementation of these solution strategies. Consequences include the survival of the patients, an improved quality of life, control over the chronic condition, empowerment, hope for other opportunities, and a contribution to empower other patients with similar chronic conditions [47,48]. Section 6 is dedicated to discussing these consequences.

### 4.4. Description of the Health Coaching Systems

Table 3 summarizes a description for the 28 different health coaching systems that were included in the primary studies. Among them, eight use the system as the main tool for the patient’s self management of the chronic disease. As an example, the “HeartCycle” application provides support for the patients to self-manage their medical, nutritional, and physical activity to avoid cardiovascular diseases repercussions [21]. In another five studies, health coaching systems are used for assisted training purposes. For instance, the “iHeart” health coaching system trains patients with hypertension to improve their cardio-respiratory endurance based on their current vital signs [52]. In two other studies, health coaching systems are used as a screening tool. The health coaching system is used to screen for the patients’ arthritis current state using the patients’ physical activity, their vital signs and other specified measures [20]. Moreover, health coaching systems are used as an educational tool to instruct patients about their chronic condition. The educational aspect of the health coaching system varies depending on the target audience. “Tako game” illustrates this concept by encapsulating a game into a health coaching system aiming to instruct young diabetes patients about their chronic condition [21]. Finally, with regards to the chronic conditions where the pressure is on mental health, health coaching systems channel their dedication to counseling their patients either by connecting patients with health coaches [30] or through automated chatbots [37].

Almost two-thirds of these studies (*n* = 19, 62%) prefer to develop the health coaching system on a stand-alone application while the remaining third uses a web-based platform to develop the health coaching system. The interaction between the health coaching systems and the patients is nuanced between the use of an expert system or involves machine learning into the interaction. The included studies cover a range of 14 different chronic condition. Table 3 summarizes these findings.

## 5. Analysis

### 5.1. Roles for Artificial Intelligence-Based Health Coaching Systems

We analyze the extracted data elements. We infer the roles that the artificial intelligence health coaching systems play in supporting patients self manage their chronic condition. The features of the AI-based HCSs are defined by the selected user stories, and specifically by the demands of the end-users (chronic patients), as they have been captured and prioritized by the product designers. Thus, these roles enable the identification of the important features impacting the end product (AI-based HCS). These features are influenced by the context of the disease, intervening and causal conditions of the patient, as well as the chronic disease itself (phenomena), as depicted in Figure 1. Following are the seven identified roles.
Develop Adherence: the aim of this role is for the health coaching system to train the patients adhere to their care program. This is achieved by coaching them especially throughout the first months of the treatment. Usually, the system achieves this role by setting a goal to the user. It also provides the user with the possible methodologies to achieve the goal set by the system [20]. Throughout the duration of the care program, the patient’s adherence to the care program is developed with the help of the system directing the user towards the right habits to practice.Inform: the inform role consists of providing medical information about a chronic disease. This information is communicated to the user in a variety of ways to support patients’ education. This leads the users to understand the cause and effect of specific actions on their chronic condition. The aim is to support the patients understand their conditions and, hence, adhere to the care program provided.Motivate: this role comes into action when users are in moments of set back or depression. This is due to the mental exhaustion their disease is putting on them. Although patients, who have already developed adherence, are informed about the benefits of the care program on their health and wellness, they may often become less motivated. In such a situation, they decide to discontinue the care program. Thus, an immediate intervention from the health coaching system to motivate them is needed. The motivate role shows the patients their past achievement in managing their condition. This is usually done by providing the users with the baseline measurements that were taken at the very beginning of the care program, and showing them how far they came ever since. The progress shown is often motivating to the patients [20,43].Remind: this role relies on notifying the users on the right time in case they forget to accomplish a task towards their care program. The remind functionalities involve the users less in terms of time spent interacting with the health coaching system as compared to the other previously described functionality roles [43].Prevent: this role relies on observing the user’s chronic condition health indicators in order to prevent accordingly a health crisis in which the patients find themselves unprepared for [41,46]. In this role, systems monitor the indicators’ thresholds or use machine learning to predict when will the next health crisis occur [40,41,46]. Thus, this methodology minimizes the risk for an unexpected crisis [40].Build a Care Network: for this role, systems rely on establishing a social bond among the patient and other users of the system. A bond is also established among stakeholders that are not involved in using the system, but are invested in the patient’s management plan. These stakeholders help the patients in coping with their chronic disease. Building a care network aims at bridging the gap between the patient and the health coaches involved in the care process of the patient. Such health coaching systems are designed to allow clinicians, caregivers, families, and friends of the patients to have access to specific parts of the patients’ data [22,47,48].Entertain: this role refers to the idea that for certain users, the chronic condition they face is managed easier in a fun interactive way. Some applications use serious games [37,38], while other rely on gamifying their platforms to engage the users and gain their time and attention [21,28]. In such systems, rewards, scores, and other engaging methods are adopted [21,28,37,38].

### 5.2. Role Combinations for Artificial Intelligence-Based Health Coaching Systems

The nature of the chronic disease has a direct influence on the design of health coaching systems (HCSs). We discuss how the combination of these roles mentioned in the previous section form the end design of AI-based HCSs of different chronic conditions. We identify six common combination categories applying a coding scheme on the 28 HCSs.
The Full Multilateral HCS combination category: is characterized by the combination of all the HCS roles in the same system. The HCSs for cardio-vascular diseases, chronic obstructive pulmonary disease, and multiple sclerosis all fall under this category.The Unilateral HCS category: is a configuration focused on a system implementing only one role. Chronic musculo-skeletal pain, Parkinson, and rare chronic diseases fall under this category.The Opposing Bilateral HCS combination category: refers to different HCS roles, if combined together might undo the progress that each role has contributed to. An example of that would be the inform and entertain functionality. In the majority of cases, developing digital games is mainly intended for entertainment. Therefore, it is rare to find a video game with the intention to inform and educate. However, combining these two functionalities is possible, as it is the case for the diabetes HCSs for adolescents [37,38].The complementing Bilateral HCS combination category: complementing bilateral HCS roles refer to the roles that complement each other. The way these roles complement one another is by one role contributing to the development of the other. The end goal is to achieve better chronic disease management. Examples of this configuration is the “remind” and “prevent”. The roles in the HCS that remind patients with chronic migraine to take medications also contribute in preventing a migraine crisis from happening [40,41]. Another example are the motivate and develop adherence roles. Incorporating those roles simultaneously in the system motivates patients to maintain their chronic disease management schedule. The maintenance of a well managed chronic condition care schedule encourages patients to adhere to their care programs.The one-role-missing combination category: is characterized by a HCS incorporating all the roles except one. The case of arthritis and obesity falls under this category.The blurred roles combination category: is characterized by some roles not being incorporated fully or accurately. The case of serious mental illness falls under this category.

Table 4 summarizes the results and findings for the HCSs and their configuration categories.

## 6. Discussion

We elaborate, in details, on the resulting role combinations for the HCSs. We use the included 31 studies as an illustration to the different combinations.

### 6.1. Multilateral Combination Category

We use the case studies of both cardiovascular diseases and multiple sclerosis shown in Table 5 to illustrate the multilateral combination category.

The difference in the number of participants for cardiovascular diseases’ studies as compared to the other chronic conditions is noticeable. While the majority of the experiments recruit participants in the range of five to 50 participants. The cardiovascular diseases studies recruited 210 participants in total. This is due to the quantitative nature of the experiment. Sideris et al. attribute this abundance in terms of participants to the fact that cardiovascular diseases are the first cause of death in the US.

In the three selected studies VeraMunoz et al., Sideris et al., Ayobi et al., the authors emphasize the importance of the process of self care and wellbeing management in addition to the clinician’s follow up. The authors argue that clinicians face a lack of control and uncertainty in managing the chronic disease when patients are not involved in the process of self-care. The authors [21,22,42] argue that the most efficient way to manage this uncertainty aspect is to prepare for it and develop proactive ways to creatively engage and develop health coaching programs. Systems for this reason help in engaging both the clinician and the patient in that process where all the roles are emphasized and encouraged. Among the existing roles in the developed HCSs are: learning about the disease (Informing), maintaining regular physical activity, a regular medicine schedule intake, and good nutritional practices (developing adherence), and caring for the patients mental well-being through a social support group (Building Care Network, and Motivating).

When patients are selfly willing to be involved in a self-care process, they creatively find effective self-tracking tools, even when HCSs are not accessible for them [42]. This observation is made by Ayobi et al., when the authors conducted a series of in-depth interviews with participants to understand which functionalities are needed most by patients. Ayobi et al. realized that patients find even a simple paper diary or a wearable fitness tracking device effective as a reminding tool (remind role).

The authors, prior to the development of a HCS, gathered the different types of engagements for the patients with both the HCS as well as the clinician recommendations. These gathered elements incentivized the design of an all inclusive multilateral system design. This design is reflected on the HCSs, as it is observed in both cardiovascular diseases and multiple sclerosis systems.

### 6.2. Unilateral Combination Category

We use the case studies of both Parkinson [46], as well as rare chronic diseases [47,48], as shown in Table 6 to illustrate the unilateral combination category.

Nunes and Fitzpatrick explain how living with parkinson disease dictates a complicated life and one full of “mundane challenges” [17]. These challenges are usually not considered by designers. Among the examples given are the complexities regarding constantly scheduling medication timetables. Patients need to prepare the medications by placing them in specific places, so as not to forget them. Patients with Parkinson also need to plan ahead of time for when to take these medications according to their activities for the day. They also have to consider for the medication’s effect time-span in order to optimize their effect per intake. This is especially important, because, if these medications are not managed well, they loose their curing power. Another aspect mentioned by Nunes and Fitzpatrick is the progressive impairment that Parkinson imposes on the patients’ abilities and, thus, makes them abandon some activities that were before the disease vital for the normal pace for the day-to-day life. According to Nunes and Fitzpatrick, the patients’ reasoning, behaviors, and personas change as they progress with the chronic disease. Patients with Parkinson become anxious as they start to live with the uncertainties that the disease hides, and they are always ready for last minutes changes of plans, as they cannot predict when will the effects of the disease take over their motor ability. This severely impacts their wellbeing.

A similar effect is observed by MacLeod et al., MacLeod et al. in both of their studies about rare chronic diseases [17]. One of the goals behind MacLeod et al.’s study is to understand the dynamics of how patients with rare diseases pass on knowledge with each others. This knowledge is niche to those specific patients with that rare condition. Even clinicians might not be knowledgeable about it due to the rarity of the disease. MacLeod et al. observe that, for rare chronic diseases, patients develop “expertise that is uniquely different from the clinicians’ expertise” [47,48]. This means that patients with rare chronic diseases take more time to study their disease and mostly share their lived experiences with the very few people that have the same condition. MacLeod et al. understand that designing HCSs for this category of patients needs a balance between what clinicians advice, and what patients themselves find in terms of how to manage their rare diseases online through their care network [47]. This balance enables the integration of a coherent system targeted for this category of patients. MacLeod et al. found through interviews that communities with rare chronic diseases are extremely active by building strong care networks of the same rare chronic disease online [47]. The care networks are usually international, as patients with the same rare chronic disease are scarce in only one country. MacLeod et al. also reveal that patients with rare chronic disease share the latest research with each other about their disease and try to consult each other about their lived experiences. Therefore, MacLeod et al. conclude that the best way to design a HCS for patients with rare chronic diseases is a system that embraces patients activity and care network online [47]. This system also needs to include clinicians in that process to verify and monitor the pieces of research circulating among the care circle. Thus, false inferences or poor decisions can be avoided.

Although there exists a difference in the nature of the chronic diseases investigated by MacLeod et al. [47], MacLeod et al. [48] and Nunes and Fitzpatrick [17], both of their developed HCSs fall under the unilateral combination category. In the unilateral configuration, the HCS focuses mainly on one role. In the case of rare chronic diseases the focus is on using the system to build a care network as the primary function, motivating and developing adherence as the secondary function with less importance given to it. For the Parkinson disease, the focus is given to developing the adherence of users. The patients need this functionality role to face the change in their physical abilities due to the disease. However, the system does not focus only on the functionalities to develop patients’ adherence, but also sheds light on the remind and prevent functionalities, as they contribute to the adherence development.

### 6.3. The Opposing Bilateral Combination Category

We use the case of two diabetes studies performed by denAkker et al. and Harris et al. shown in Table 7 to illustrate the opposite bilateral combination category.

Both of the studies design HCSs for diabetic adolescents. They suggest that clinicians and health coaches fail in motivating and engaging adolescents in their chronic disease care program, especially when an overwhelming amount of information is communicated to the adolescents. Harris et al. observe that teenagers, in general, and the ones with diabetes have a rebellious persona towards receiving information with regards to their chronic disease. Adolescent patients see this amount of information as “overwhelming” requiring them to adhere at an early age to a rigorous care program. Therefore, the authors [37] realized that an effective way to improve accessibility and practicality of health coaching strategies for this part of the population is to design systems that are appealing to the preference needs of the youth. Harris et al. realize that the best way to do it is through a mobile “serious game”. The term serious game refers to a game with the purpose of entertaining and educating at the same time. The authors explain that adolescents nowadays have a natural tendency to effortlessly use a smartphone. Therefore, operating an application or playing a mobile game is done naturally and intuitively. Designing a health coaching platform, according to denAkker et al., requires maintaining adolescents’ preferences in mobile games. These preferences are mainly the fun and entertainment aspects to the game, while at the same time involving the information that needs to be transferred to the adolescent patients. Harris et al. refer to storytelling, character personification, point scores, and other techniques to make the designed game resonate to adolescents. However, the authors keep enough information about the adolescent patients’ diabetes health status through constant readings transmitted by the sensors connected to the system. Harris et al. and denAkker et al. successfully involved two opposite roles, informing about the chronic disease, without trading off the fun aspect in the system to make it appealing to its target group. These two roles, inform and entertain, although contradictory to one another, are equally represented in the HCS.

### 6.4. Complementing Bilateral Combination Category

We use the case studies of two migraine experiments performed by Park and Chen and Schroeder et al. shown in the Table 8, below, to illustrate the complementing bilateral combination category.

Schroeder et al. in their study explain that chronic migraine is a chronic disease unlike other chronic diseases. It “is characterized by unpredictable, intermittent, and poorly understood symptoms”. Therefore, the health coaching care program varies widely from person to another, and from clinician to another. This entails that each patient might track a wide array of different symptoms that relate to their unique chronic migraine case. For those reasons, Schroeder et al. conclude that, in order to contribute to reducing the frequency and intensity of the symptoms, there needs to be a deep investigation about how people with migraine track and use the data related to their disease. Therefore, the study that the authors conduct involves 279 different participants. The investigations use in depth interviews to understand the different ways that patients feel and deal with chronic migraine. The investigation also consists of patients self documenting the tools, as well as the techniques they use to deal with the disease. Schroeder et al. conclude that the best HCS to enhance patients with chronic migraine’s lives is a self-tracking system that helps identify personal triggers to causing intense headaches. Therefore, Schroeder et al. designed a system that was powered by machine learning where self-tracked migraine data are used to enable prediction, communication ahead of time, and prevention for the symptoms that might cause an eventual migraine crisis. In this complementing bilateral category, each of the two role functionalities complement one another. The two roles here are the prevent and remind.

### 6.5. The One-Role-Missing Combination Category

We use the case studies of arthritis and obesity performed by Gupta et al. and Barnett et al., as shown in Table 9, below, in order to illustrate the one role missing combination category.

When it comes to the one role missing category, it is important to understand why among all available seven roles, one is not considered. In the studies included for the analysis, both the arthritis study done by Gupta et al. and the obesity study conducted by Barnett et al., discarded one role in the development of their HCSs. The obesity study discarded the prevention role, while the arthritis study discarded the entertainment role. For obesity, and according to Barnett et al., it is one of the chronic diseases that has a methodical care program, where the results are to be calculated and predicted. Therefore, unexpected results or symptoms rarely occur. The feature set for the HCS remains effective and sufficient, even without a prevention role. With regards to the arthritis study, the same rationale is behind, but, for this case, the HCS not involving the entertainment aspect instead of the prevention aspect. The reason for this decision, according to Gupta et al., is that the system focuses on involving vizualization heavily, since the physiotherapist coach needs to monitor several parameters at a time to design the most effective coaching program in real time. Gupta et al. see that the system is able to skip presenting the information in a fun entertaining way due to the fact that the system is directed to the therapist coach in the first place.

### 6.6. The Blurred Roles Category

We use the study of serious mental illness performed by Aschbrenner et al., as shown in Table 10, below, to illustrate the blurred roles combination category.

In our analysis, the blurred roles category is represented by the serious mental illness study that was conducted by Aschbrenner et al. As the study involved clinical research, there is an observation that patients with serious mental illness, such as schizophrenia and bipolar disorder, have a higher risk of early death due to cardiovascular diseases and other preventable chronic illnesses. Aschbrenner et al. pointed that young adulthood is an important opportunity for the development of health coaching lifestyle interventions to improve the long-term health and quality of life in this fragile population. The authors also mention that serious mental illnesses cause other chronic diseases if not addressed as early as possible. When different chronic diseases get discordantly involved in a patient, treatments become challenging. One treatment for a specific chronic disease can awake symptoms of another chronic disease. Therefore, Aschbrenner et al.’s study compared two HCSs that were targeted to different treatments: the peerFit and BEAT interventions. The first is targeted in order to develop adherence to treat the cardiovascular disease which is a consequence of the serious mental illness repercussions. The second is meant to develop a care network through the HCS to help patients with serious mental illnesses to develop a social circle. Managing two chronic diseases in parallel creates a dispersion of focus in the overall design of the HCSs, and thus the blurred roles reflected in these studies.

## 7. Conclusions

### 7.1. Concluding Remarks

Across the primary studies, the authors of the selected articles proposed numerous health coaching systems. These systems vary in terms of the roles they play in order to support the patient through the self-management of the disease. Across the subset of the analyzed studies, there were no common efficiency metrics in use to measure the effectiveness of the HCSs. Hence, the diversity of the roles and their combinations that these systems play. This is observed first-hand across the primary studies, as each study attempted to measure the outcome of its system differently. Moreover, the nuanced study designs across the primary studies adds to the complexity of proposing a unifying effectiveness metrics. However, this does not prohibit drawing conclusions at the study characteristics level instead of a unifying global level. The following paragraphs elucidate the main findings and the conclusions drawn at the characteristic level.
The study measured outcome: Table 2 and Table 3 revealed the measured outcome of each primary study. Throughout our analysis, we sorted these outcomes across four categories; the acceptability of the system by the participants, its effectiveness on their self-management of the disease, its usability, and finally the adoption of the participants for the system post the experiment phase. It is noteworthy that studies often do not explicitly declare their intended outcome. Therefore, the classification made across Table 2 and Table 3 is subjective to the understanding of the authors of this work. In the unlikely event when the authors of the primary studies declare the measured outcome at the end of the experiment phase, objective metrics to measure it are set before the start of the experiment. These metrics are usually represented by a question with a binary answer (e.g., Did the participant adopt the system past the experiment phase or not?). This is observed in the four studies that measure the adoption of the system post the experiment phase [22,32,36,48]. At the opposite end of the spectrum, the majority of the studies that do not specify the measured outcome metrics, as illustrated by the studies that measure the effectiveness of the system (refer to the full list of the 17 studies that fall under this category in Table 3), find the task of providing a trenchant decision with regards to the outcome of the study difficult. This may be due to different causes. Among these causes are: the sample of participants involved in the study, as well as its setting. These two causes are developed in the two points below.Sample of participants. The sample size across the primary studies ranged between five up to 375 participants [28,35]. This is a significant difference among studies that measure the outcome of systems with multiple similarities. This difference in the number of participants involved in each experiment changes the dynamics of measuring the outcome out of the study. For the study with only five participants [28], it is logistically, and technically easier to measure whether the patients adopted the system past the experimentation phase. However, when the number of participant is 55 folds larger, with a number of participants of 275 [35], finding the objectivity in measuring the effectiveness of the system on the participants is supposedly significantly difficult. The number of participants involved in the study may not be the only factor hindering an objective measurement outcome of the health coaching system. The following point discusses another factor that may be involved in this predicament.Setting of the study. The setting where the study is conducted may also be a factor in hindering an objective measurement for the effectiveness of the health coaching system. Table 2 shows that the setting of the primary studies varied between a clinical, a community, as well as an academic setting. These settings, depending on the health coaching system, may either involve favoritism, impartiality, or dis-favoritism to the approval of the participant for the system. Allowing the freedom for the participants to experiment with the system in their home setting is less stressing when compared to testing the system in an academic or a medical setting.

The three points mentioned above do not enclose the totality of the factors involved in the success or failure of a health coaching system. However, these points are the remarks that the authors find noteworthy to mention before initiating the design and implementation of a study involving a health coaching system.

### 7.2. Limitations

The authors of the primary studies revealed some limitations that their conducted studies suffer from. Most of the noted limitations correlate with the observations made in the concluding remarks. We discuss two types of limitations. Limitations at the level of the primary studies’ design and limitations at the level of the conducted literature review. The below two sub-sections expand and discuss these two types.

#### 7.2.1. Limitations at the Level of the Conducted Studies

The limitations at the level of the conducted primary studies cover the study design, the number of recruited participants, and the duration of the study. The authors of the primary studies acknowledge that designing and conducting a feasible study is a trade-off between these three parameters mentioned [35] against the invisible parameter of the budget allocated to conducting the study. Studies with large numbers of participants (e.g., 375 participants as reported by Fadhil et al.) often require large budgets and a bigger number of researchers managing and administrating the study itself. Moreover, Fadhil et al. acknowledge that conducting the study in the home setting of the participants instead of the clinical setting would increase the significance of the results with regards to the effectiveness of the system. This is especially pertinent, because the health coaching system designed is for depressed patients. Additionally, Fadhil et al. acknowledge that the choice of the metrics in order to measure the effectiveness of the system is difficult, since depression is a chronic condition that is difficult to objectively quantify. The large number of participants involved in the study also adds to the difficulty of the quantification of the effectiveness of the system.

In contrast with Fadhilet al.’s study, Mateevitsi et al. chose a different approach as the number of participants is significantly lower (five instead of 375). This smaller sample size has its advantages in terms of: following up with the participants, allowing a more comfortable home setting instead of a clinical or academic setting, and in terms of the choice and control of the evaluation metrics. Moreover, the smaller number of participants allowed for a follow-up period of six months past the end date of the study. However, the limitation that the smaller number of participants brings is the lack of objectivity in evaluating the health coaching system. Claiming that five out of five participants adopted the use of the system does not translate into claiming the adoption of the system by a larger population size with different backgrounds and chronic condition needs. Therefore, a small number of participants is a limitation that may discredit the significance of the results obtained after experimentation.

Lastly, the final element that may hinder the significance of the results and cause bias for the conducted study is its duration. The duration of the primary studies ranged between one week up to 40 weeks [33,43]. The studies conducted over a longer duration allow for more flexibility and experimentation for the participants. However, the limitation that studies suffer from when the experiment is conducted over a shorter period is the lack of flexibility and time for the participants to become familiar with the system. Moreover, the data collected over few weeks are scarce when compared to the data measured over several months. This has a preponderant weight over the significance of the results reported over the studies.

#### 7.2.2. Limitations at the Level of the Literature Review

We list the limitations that this work suffers from at the literature review level. We also explain the trade-off accepted by the authors that justify these limitations.
The methodology followed to perform the literature review: the authors follow a grounded theory literature review methodology. This methodology is more common in the social sciences compared to computer sciences [15]. This is because it allows the construction of theories through methodical gathering and analysis of data. It also uses inductive reasoning, in contrast to the deductive model based on hypothesis of the scientific method. Given the fields that are involved in this review (HCI, AI-based HCSs, chronic diseases), it is also appropriate to follow the systematic review methodology. However, the authors referred to GTLR to enable the identification of the different features that characterize artificial intelligence based health coaching systems used by patients with chronic diseases. This approach treats the content of the primary studies as empirical data. This led to different codes to emerge. The process also involved revising these codes, and grouping them into themes. This subsequently led to grounding the roles that the artificial intelligence health coaching systems play, as well as grounding the the combination of the roles that form the design of AI-based HCSs for different chronic conditions.The search and selection process: while there exists some commonalities among the search and selection strategies of both the systematic review and the GTLR methodologies. These commonalities are summarized in a systematic approach to searching, analysing, and reporting a set of literature (e.g., The screening steps, the clarification of what items are included in each step, two independent reviewers screening, a third solving inter-rater disagreements etc.) [15]. The systematic review methodology is characterized by unique and distinct guidelines to accurately present each step before reaching to the set of included primary studies. Among these guidelines that are not performed in this review: The PRISMA-style flowchart, the formal assessment of each study’s quality before including it in the set the primary studies, and the meta-analysis assessing the statistical heterogeneity among included studies.

### 7.3. Future Works

The conclusions from the analysis, discussion, and conclusion sections will be carried out to design a study that uses an AI-based HCS dedicated for supporting patients with cardiovascular diseases independently interpret their electrocardiograms (ECGs). With the advent of wearable technologies that are capable of measuring patients’ ECGs, supporting patients understand and properly read their electrocardiogram is of paramount importance [4,8]. However, this task requires a deep understanding in both the computational and medical fields. It also requires attention with regards to how the system translates the information to the users. The realization of this work unveiled the interlinking relationship between the medical field, the computational field, and the interaction between the two fields thanks to the computational sub-domain known as human computer interaction. This grounded theory-based literature review opened a window for contemplation about the ways the computational domain is of service to the medical domain.

## Figures and Tables

**Figure 1 healthcare-08-00313-f001:**
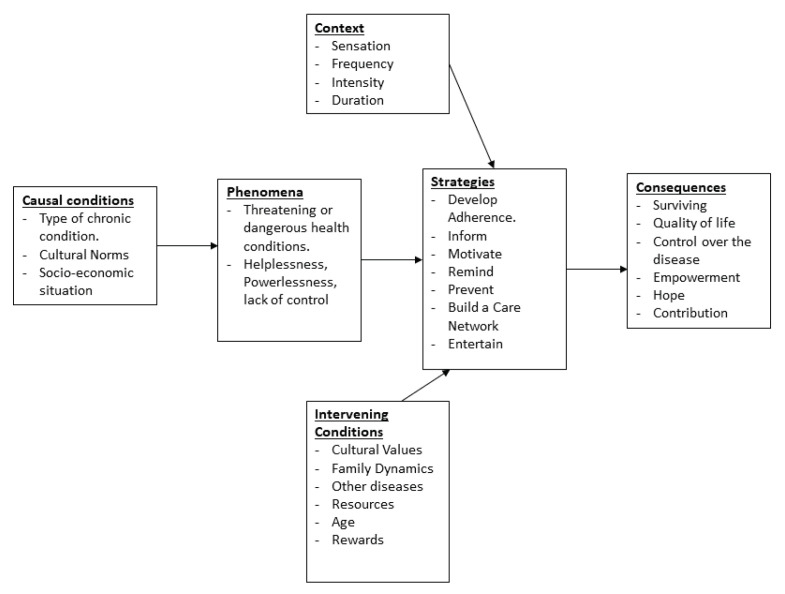
Axial Coding for the elements leading towards an innovative HCS.

**Table 1 healthcare-08-00313-t001:** Overview of the 31 papers included in the review.

Reference	Chronic Condition	Patient Profile	Technology	Number of Weeks	Number of Participants
[20]	Arthritis	Patients with mobility limitation who prefer to not be dependent on a caregiver	“FitViz” web application allows the patients to autonomously manage their physical activity trends by showing visualizations of their activity goals.	4	20
[21]	Cardio-Vascular disease	Patients with a sedentary lifestyle and difficulty in initiating the care treatment	“HeartCycle” is personal health system based on a goal oriented design and an education and coaching strategy.	13	120
[22]	Cardio-Vascular Disease	Patients who live an unsane lifestyle and do not contemplate changing it. This lifestyle consists of a junk food based diet, and sedentary living	Remote health monitoring system based on customized messages sent to patients. The system also uses machine learning to predict the following week’s patient adherence.	26	90
[23]	Cardio-Vascular Disease	Ageing population with recurrent and unpredictable heart failures	System with predictive, preventive and personalized medicine approach where patients are leading their management, supported by an easily accessible online application that takes advantage of artificial intelligence	8	90
[24]	Cardio-Vascular Disease	Patients with abrupt cardiac failures who need to be monitored all the time for prevention and precaution of the next crisis	Portable ECG device that provides captured ECG data and suspected waveform to identify sporadic and chronic events of heart diseases	-	-
[25]	Cardio-Vascular Disease	Patients who find a difficulty in maintain a good balance between work, social life, a healthy diet or a regular medicine intake	“The eMate intelligent coaching system” uses the computerized behavior intervention model to send the user tailored information to motivate behavior change and overcome the patient’s lifestyle challenges.	12	40
[26]	Cardio-Vascular Disease	Patients with abrupt cardiac failures who need to be monitored all the time for prevention and precaution of the next crisis	Hapicare, a system that applies ontology-based uncertain reasoning over IoT sensors data and self-assessment to prevent the patient’s heart crisis	-	-
[27]	Chronic Neck Pain	Patients on treatment for neck or shoulder pain who need to constantly track EMG patterns of the Trapezius muscles in order to estimate their level of relaxation.	Personal Coaching Systems (PCS) that uses on-body sensing, combined with smart reasoning and context-aware feedback to support users in developing and maintaining a healthier behavior	-	-
[28]	Chronic Musculo-skeletal Pain (CMSP)	Patients with CMSP who are depressed and unproductive in their offices	“Healthbar” is a persuasive ambiant dislplay mounted on the office computer screens of workers to remind them to perform some in-office physical activity.	4	5
[29]	CMSP	Patients who associate painful memories and negative experiences related to CMSP	“PMP: Pain Management Program” system to coach patients, tailored by machine learning and physiotherapists.	10	9
[30]	CMSP	Old patients with CMSP who suffer from a limited ability to move	Clinician-in-the-loop (CIL) visual interface that provides clinicians with patient behavior patterns derived from smart home data.	10	60
[31]	Chronic Obstructive Pulmonary Disease	Patients suffering from breathlessness and fatigue. This obstacle is making physical activity challenging for them	Use of machine learning to predict the health decline of the pulmonary disease before the problem occurs.	4	9
[32]	Chronic Obstructive Pulmonary Disease	Breathlessness and fatigue, making physical activity challenging	Compared the use of patients for three different prototype mobile applications. Each one uses a type of persuasive technology design principles, as defined by the persuasive systems design (PSD).	12	87
[33]	Chronic Obstructive Pulmonary Disease	Patients who are frustrated due to impaired performance of daily living activities	System that trains a predictive algorithm to prevent exacerbations. The system also provide personalized virtual coaching to improve medication adherence and activity level.	40	10
[34]	Depression	Patients who are unmotivated, depressed, and antisocial	Use of a variety of technology channels to connect with others patients to express their moods. System that creates solutions and challenges to overcome depression.	4	20
[35]	Depression	Depressed patients who face low physical activity, harmful unhealthy diet, and high levels of stress	human–virtual agent mediated system that leverages the conversational aspect to handle menial caregiver’s works by engaging patients in a conversation.	4	375
[36]	Hypertension	Patients who need to maintain a continuous check of their vital signs to prevent health repercussions	“iHealth365” is an interactive system of virtual healthcare assistant to help patients easily understand their health conditions, and then well manage it for a better wellbeing. It analyzes the result of regular physical examination to evaluate the health risk and provide personalized healthcare services for patients	30	10
[37]	Diabetes	Adolescent patients who lack the understanding, importance and motivation to manage diabetes	Integrated Personalized mobile e-Health Coach for adolescents to support them in dealing with diabetes.	5	16
[38]	Diabetes	Adolescents patients who lack the understanding and motivation to manage diabetes	Platform that uses gaming and coaching for adolescent patients with type 1 diabetes	2	18
[39]	Diabetes	Patients who are unable to manage their diabetes by themselves	System that attempts to visualize personalized blood glucose forecasts. These forecasts promote diabetes self-management, as well as understanding key styles and visual features that resonate with patients	4	13
[40]	Migraine	Patients who are scared of unpredictable, intermittent breakouts in everyday life	Application to track a wide spectrum of life events across intermittent time stamps. The application therefore helps patients make sense of subjective information.	2	12
[41]	Migraine	Patients who have unpredictable, and poorly-understood pain, sensitivity to light, and impaired cognition.	System to track the nature of the symptoms to accommodate the uncertainty of migraine pain.	-	279
[42]	Multiple sclerosis	Patients who are Passionate about pursuing a healthy lifestyle. They are engaged in self-tracking practices of their own volition	System that adopts self-care practices with different self-tracking technologies.	-	15
[43]	Obesity	Patients with low motivation. They find a difficulty in initiating their care program.	“myPace” Obesity management system that connects dietitians to patients. It shows to the patients their regular progress updates as well as tailored advice provided by dietitians.	1	10
[44]	Obesity	Patients who are unconscious about their nutritional choices, or are unmotivated to change their behavior towards a sane and healthy lifestyle	e-coaching system that draws from the techniques of professional nutritional coaches	1	10
[45]	Obesity	Patients who suffer from unhealthy dietary habits limited nutrition literacy and food literacy skills	Serious game aiming at building nutrition literacy and food literacy skills in adolescents and young adults.	4	10
[46]	Parkinson	Patients living with Parkinson’s perform multiple activities in a mundane way, surprisingly different from people who do not manage this condition	Technology that adopts simple reminder methodologies to foster the mundane care for Parkinson for a better wellbeing	-	-
[47]	Rare Chronic Diseases	Patients appropriate the disease to their identity	social support care network to find fellow patients with the same rare chronic disease	-	-
[48]	Rare Chronic Diseases	Patients who are depressed, anxious, worried about the future and lonely	Online support group and repository for updated research about the disease	22	11
[49]	Mental Illness	Patients with low self-esteem, and low confidence. They are mostly overweight obese as a consequence of a sedentary lifestyle	Two system models are proposed. 1. a mobile health group life-style intervention (PeerFIT) 2. one-on-one wearable supported mobile life-style coaching (BEAT)	…	144
[50]	Mental Illness	Patients are hesitant and hostile towards engaging in conversations with the social surrounding.	Conversational health coaching system with multiple interaction modalities to suit the preferences of the patient (emoji-based, written, conversational)	1	58

**Table 2 healthcare-08-00313-t002:** Overall characteristics of the included studies.

Characteristics	Number of Studies
Study Design	Quasi-Experiment: *n* = 14, Survey *n* = 13, Randomized Controlled Trial *n* = 4
Type of publication	Conference: *n* = 21, Journal: *n* = 9, Thesis: *n* = 1
Country	The Netherlands: *n* = 5, US: *n* = 12, UK: *n* = 4, Italy: *n* = 3, Canada *n* = 1, Spain: *n* = 1, Switzerland: *n* = 1, Austria: *n* = 1, France: *n* = 1, Taiwan: *n* = 1, Greece: *n* = 1
Years of publication	2019: *n* = 9, 2018: *n* = 6, 2017: *n* = 4, 2016: *n* = 1, 2015: *n* = 5, 2014: *n* = 3, 2013: *n* = 2, 2012: *n* = 1
Sample size	<50: *n* = 19 50–100: *n* = 8, 100–375: *n* = 4
Mean age	34.6 years
Sample type	Non clinical: *n* = 21 Clinical *n* = 10
Setting	Clinical *n* = 9, Community *n* = 17, Academic *n* = 5
Measured outcome	Acceptability: *n* = 5 Effectiveness: *n* = 17 Usability: *n* = 5 Adoption: *n* = 4

**Table 3 healthcare-08-00313-t003:** Summary description of the different health coaching systems.

Category	Characteristics	Reference
Purpose	Training	[25,27,32,36,49]
	Screening	[20,30]
	Self-Management	[17,21,22,23,26,31,34,42,44]
	Counseling	[29,35,37]
	Education	[28,38,39,47,48,50]
	Diagnosing	[24,33,40,41,43,45]
Platform	Stand alone software	[20,21,23,28,29,30,32,33,35,36,37,38,39,40,42,44,45,49,50]
	web-based	[17,22,24,25,26,27,31,33,41,43,47,48]
Interaction	rule-based	[17,20,28,29,33,34,37,38,39,40,42,44,45,47,48,49]
	artificial intelligence	[21,22,23,24,25,26,27,30,31,32,35,41,42,43,50]
Chronic condition	Artritis	[20]
	Cardio vascular diseases	[21,22,23,24,25,26]
	Chronic Obstructive Pulmonary Disease	[31,32,33]
	Chronic Muskulo Skeletal Pain	[28,29,30]
	Diabetes	[37,38,39]
	Depression	[34,35]
	Migraine	[40,41]
	Multiple Sclerosis	[42]
	Obesity	[43,44,45]
	Parkinson	[17]
	Rare Chronic Diseases	[47,48]
	Mental Illness	[49,50]
	Chronic Neck Pain	[27]
	Hypertension	[52]
Measured outcome	Acceptability	[25,28,37,38,39]
	Effectiveness	[21,23,24,26,27,29,30,31,33,34,35,42,43,44,45,46,50]
	Usability	[20,40,41,47,49]
	Adoption	[22,32,36,48]

**Table 4 healthcare-08-00313-t004:** Summary of the category results along with their figures and reference studies.

Category	Chronic Condition	Reference
Multilateral		[21]
		[22]
	Cardio-Vascular	[23]
	Diseases	[24]
		[26]
		[25]
	Chronic	[31]
	Obstructive	[32]
	Pulonary Disease	[33]
	Multiple Sclerosis	[42]
Unilateral	Chronic Muskulo	[28]
	Skeletal Pain	[30]
		[29]
	Parkinson	[46]
	Rare Chronic	[47]
	Diseases	[48]
Opposing Bilateral	Diabetes	[37]
		[38]
		[39]
Complementing Bilateral	Migraine	[40]
		[41]
	Chronic Neck Pain	[52]
	Depression	[34]
		[35]
One-role-missing	Arthritis	[20]
		[43]
	Obesesity	[44]
		[45]
Blurred Roles	Serious Mental Illness	[49]

**Table 5 healthcare-08-00313-t005:** Multilateral combination category category chosen examples.

Category	Chronic Condition	Reference
Multilateral	Cardio-Vascular	[21]
	Diseases	[22]
	Multiple Sclerosis	[42]

**Table 6 healthcare-08-00313-t006:** Unilateral category chosen examples.

Category	Chronic Condition	Reference
Unilateral	Parkinson	[46]
	Rare Chronic	[47]
	Diseases	[48]

**Table 7 healthcare-08-00313-t007:** Opposing bilateral chosen examples.

Category	Chronic Condition	Reference
Opposing bilateral	Diabetes	[37]
		[38]

**Table 8 healthcare-08-00313-t008:** Complementing bilateral chosen examples.

Category	Chronic Condition	Reference
Complementing bilateral	Migraine	[40]
		[41]

**Table 9 healthcare-08-00313-t009:** One-role-missing chosen examples.

Category	Chronic Condition	Reference
One-role-missing	Arthritis	[20]
	Obsesity	[43]

**Table 10 healthcare-08-00313-t010:** Distorted category chosen examples.

Category	Chronic Condition	Reference
Distorted	Serious Mental Illness	[49]

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
