# Peer review of "On How Chronic Conditions Affect the Patient-AI Interaction: A Literature Review"

_healthcare, 2020, doi:10.3390/healthcare8030313_

Round 1

Reviewer 1 Report

Dear authors, 

first of all congratulations with an interesting topic and the GTLR methodology you have selected for a systematic literature review.

Below you will find some methodological issues and a few minor terminology issues for your consideration.

1) aim - objective - motivation - methodology

objective is formulated as to provide and overview of the features characterizing AI based health coaching systems". The main question of this work - "which criteria taken into consideration by the designers of AI based coaching systems during the design process" and then you formulate the fundamental research question of your paper -- what is the influence of different chronic disease on the design choices for AI-based coaching systems. 

It looks from the list of the selected papers (and their corresponding applications) that you have covered both AI and expert systems in equal proportions, which is not in line with your scope definition. Therefore you shall either reformulate it in your objective-motivation-key question or modify your analysis, by excluding expert systems, which are not subset of AI.

2) usage of GT:

one of the steps which looks to be not covered (at least not described) is axial coding, specifically interrelations between categories are not analyzed, which would provide useful insights.

3) as a result of thorough analysis and induction of identified categories, you came up with 7 roles, which according to you reflect the design decision influenced by different chronic diseases (lines 236-240). However, this statement is not reasoned or illustrated and is not obvious. 

Your inducted roles and their combinations can be translated to the main feature sets in software. Speaking in the terms of CS, features are defined by the selected user stories, and specifically by the demands of the end-users (chronic patients) as they have been captured and prioritized by the product creators. If that was your meaning, could you please reformulate.  Otherwise, the term "design decision" influenced by chronic disease sounds misleading.

4) The defined 7 roles and their grouping may be highly subjective. One may say that "Remind" is a subgroup of "Motivate", develop adherence is a subset of Inform. Surely, this subjectivity is fully in line with the principles of quality research and in particular GT, but your next analysis of the existing role combinations looks less reliable due to the reasons explained above. I would recommend to consider that in your next research.

5) the combination category opposing bilateral looks superficial. Event the example with games (why limit to video games??), actually frequently is used for educational purposes, especially in pediatrics. take diabetes type I. as you rightly said the combination is still possible, and in such case that becomes complementing bilateral category. Here goes the question, why in your example Diabetes case is identified as opposing bilateral, although it looks to be complementing, i.e. entertaintment combined with education? 

6) Your central conclusion statement "The diversity of these roles ...dictate the impossibility of proposing a unifying metrics ..." This conclusion is not supported or based on the findings of your research. It is even counter-intuitive: there are plenty of cases when for wide variety of entities, it is possible to find unifying metrics, take drugs for example, take medical software devices, and finally,  take AI algorithms for healthcare, being evaluated and measured by FDA protocols. Instead, you may state that in the subset of the analyzed studies, there were no common efficiency metrics in use. 

  Terminology:

  • "paradigm shifts" of healthcare is mentioned numeruos times in your paper. However, the listed phenomena is mostly change of certain characteristics (or transitions) of healthcare service provision, e.g. patient focus, holistic approach, collaborative treatment, etc. These are all natural changes which we experience and research in healthcare, and which eventually will bring (or not) to a certain paradigm shift. however, stating that each of them is a healthcare paradigm shift is an overstatement.  
  • line 85, what do you mean exactly by "end-design", please be more explicit. 
  • line 232 "rule-based ontology", do you mean expert system? as all ontology are in way deterministic and rule based...it's not clear what you meant.
  • line 386 "HCS configuration" or you call it "combination category", i.e. in Table 4? 
  • line 451 "the design for the HCS remains effective ...". normally in CS we do not speak about effectivity of software design in terms of its supporting feature sets, or as you call them - the usage roles. Perhaps, design -> feature set.  

Language and other:

line 237 "generate" -> perhaps induce or infer....?

line 392 "In both cases, little attention ....to the disease." this sentence reflects trivial statement and could be omitted.

line 406 "serious game". Which game is serious? 

line 600 "unveiled to the authors the dynamics between ..." not clear what do you mean. pls. rephrase....

Author Response

We thank Dr. Reviewer 1 for their sincere review job, and for going thoroughly through the whole manuscript. Your well-put and on point feedback contributed to our learning first, and to improving the manuscript second.

We hope that the attached PDF document containing our response, as well as the modifications made on the updated manuscript will meet your exigent suggestions. 

The authors.

Reviewer 2 Report

This paper presents a grounded theory literature review aiming at providing helpful information to build health coaching systems for patients with chronic conditions. The motivation of the authors in this paper is useful; however, this current version does not satisfy me due to the following problems:

  • I expected the authors to give more information related to AI. It was not clear to me about the importance of AI towards health coaching systems in dealing with chronic conditions.
  • The number of reviewed papers is not adequate, with some of them coming from the same authors. In addition, Table 1 summarised various technologies from the literature review; however, the authors didn’t mention more in-depth these technologies afterwards. This will be a significant contribution of this paper.
  • Section 5 and 6, in my opinion, are more like a combination of background and related work than analysis and discussion. Besides, I couldn’t find out how the authors selected only 2-3 studies for their discussion in Section 6.

Minor issues

  • The authors should modify the structure of the paper. Section 2 is too short to stay individually.
  • The title of Section 4 should be changed.
  • References are still missing information (e.g., page number) and having inconsistent formats.

Author Response

We thank Dr. Reviewer 2 for their constructive feedback.

Attached is a PDF containing our response.

Sincerely,

The authors

Reviewer 3 Report

This paper provides an overview of the features that characterize artificial intelligence-based health coaching systems used by patients with chronic diseases.

The study is thorough, and the contribution is nice. I suggest addressing the following points to improve the quality of the work:

- I would like to see a paragraph discussing the latest advancements in the domain of data analytics and big data, which are the backbone of the considered topic. Specifically, I would see a discussion about the following articles (and others):

    • DARM: a privacy-preserving approach for distributed association rules mining on horizontally-partitioned data. In Proceedings of the 18th International Database Engineering & Applications Symposium(pp. 1-8), 2014.
    • BigTrustScheduling: Trust-aware big data task scheduling approach in cloud computing environments. Future Generation Computer Systems110, 1079-1097, 2020.
    • A comprehensive analysis of healthcare big data management, analytics and scientific programming. IEEE Access, 2020.

- The English in the paper needs some revision at some places.

Author Response

We thank Reviewer 3 for their constructive feedback.

Attached is a PDF containing our response.

Sincerely,

The Authors

Round 2

Reviewer 1 Report

Well done, success with your further research!

Reviewer 2 Report

The authors answered and changed considerably the content that justifies underlining doubts that are now almost clarified. I, therefore, propose the acceptance of this current version.

Reviewer 3 Report

The authors have adequately answered my comments